# Predictors of Health-Seeking Behavior for Fever Cases among Caregivers of Under-Five Children in Malaria-Endemic Area of Imo State, Nigeria

**DOI:** 10.3390/ijerph16193752

**Published:** 2019-10-04

**Authors:** Sampson Emilia Oluchi, Rosliza Abdul Manaf, Suriani Ismail, Theophilus Kachidelu Udeani

**Affiliations:** 1Department of Community Health, Faculty of Medicine and Health Sciences, University Putra Malaysia, UPM Serdang 43400, Selangor Darul Ehsan, Malaysia; oluzube@yahoo.com (S.E.O.); si_suriani@upm.edu.my (S.I.); 2Department of Medical Laboratory Sciences, Faculty of Health Sciences & Technology College of Medicine, University of Nigeria Enugu Campus, Enugu 410001, Nigeria; theophilus.udeani@unn.edu.ng

**Keywords:** health-seeking, behavior, malaria, under-five children, Nigeria

## Abstract

Fever is one of the most common symptoms of pediatric illnesses; it is an important early symptom of malaria. Fever had served as the entry point for presumptive treatment of malaria among children in Nigerian. Appropriate HSB is important when seeking treatment for fever among under-five children; this will help for better prognosis because treatment will be initiated early. This study attempted to identify caregiver’s HSB for under-five children with fever. A cross-sectional study was conducted in Imo-State, Nigeria. Appropriate HSB was operationally defined as seeking treatment from health facility within 24 h of fever. Data were obtained using pretested self-administered questionnaire. Data were analyzed using SPSS version 22. Simple and multiple logistic regression were used to determine predictors of appropriate HSB. A total of 559 eligible respondents were recruited; 103 (18.6%) caregivers had appropriate HSB. The predictors of HSB are being male child (aOR = 2.760; 95% CI:1.536–4.958), the age of child younger than 27 months (aOR = 2.804; 95% CI:1.485–5.295), employed caregivers (aOR = 1.882; 95% CI:1.014–3.493), number of household members (aOR = 2.504; 95% CI:1.464–4.283), and caregivers who decided to seek treatment at early stage (aOR = 7.060; 95% CI:1.616–30.852). Only 18.6% caregivers practiced appropriate HSB for fever cases among under-five children. It is essential to educate caregivers and emphasise on early treatment of fever and appropriate use of health facilities for fever. The findings will be used to improve intervention at the community level and will be compared with follow-up data to evaluate their effectiveness.

## 1. Introduction

Fever is the commonest symptom in children under the age of five years. Fever indicates systemic inflammations; malaria; response to a viral, bacterial, and noninfectious etiology are less common among children under the age of five years. In malaria-endemic countries, most healthcare practitioners presumed that malaria was the cause of fever; the proportion of fever due to malaria was high in the early 1990s, and the priority was to decrease malaria’s death toll [1]. The 2010 WHO guidelines for the treatment of malaria recommended that in malaria-endemic areas, clinical diagnosis of malaria should be based on a history of fever in the prior 24 h or the presence of anemia which includes pallor of the palms because it seems to be the most reliable sign in young children [2].

Malaria affects millions of individuals worldwide yearly, and the disease is a global health problem [3]. In 2016, there were an estimated of 216 million cases of malaria reported from 91 countries; it was an increase of five million cases when compared to the previous year. In 2016 the worldwide malaria mortality reached 445,000, compared to estimated 446,000 deaths reported in 2015. World Health Organization African region reported 91% of world malaria mortality in 2016 and Southeast Asia Region reported 6% of malaria mortality. WHO African region remains the highest burden of malaria cases and deaths, about 90% globally [4].

Malaria is preventable and treatable; efforts are intensely propagated in reducing the malaria burden globally. Appropriate treatment of malaria within 24 h of onset of fever might decrease malaria illness [3]. Information on health seeking behavior (HSB) for malaria treatment, the hindrance to treatment, and the use of anti-malaria drugs are important [5]. Appropriate HSB is important when seeking treatment for fever among under-five children; this could help for better prognosis because treatment will be initiated early. As suggested by WHO guidelines for the treatment of malaria, early HSB is a critical behavior that is helpful for the achievement and sustainability of controlling malaria [6].

Malaria prevalence among children 6 to 59 months in the six regions of Nigeria is as follows: In South West Nigeria it has the highest prevalence with 50.3%; in North Central Nigeria it has a prevalence of 49.4%; in the North West region it has a prevalence of 48.2%; in the South-South region 32.2%; in the North East region it has a prevalence of 30.9%; and in the South East region of Nigeria, which is the study area, malaria has a prevalence of 27.6 % in 2011 among under-five years children. Malaria accounts for 30% of hospitalizations among children under the age of five years in Nigeria [7]. In Nigeria malaria is responsible for 25% of infant mortalities and 30% of under-five mortality resulting in about 300,000 childhood deaths yearly [8]. There are consequences of malaria when treatment is delayed. Delay in seeking treatment for a child with malaria deteriorates the child’s condition, with high fever, loss of appetite and refusal to eat, and vomiting. These symptoms require urgent healthcare consultation. However, when these symptoms occur, delay or not taking the child to a health facility may lead to mortality [3]. Treatment within 24 h of onset of symptoms prevents progression of malaria [9]. Federal Ministry of Health (FMOH) and National Malaria Control Programme (NMCP) provide malaria control by preventing malaria transmission through integrated vector management, early diagnosis, and appropriate treatment of all levels of clinical cases and in health sectors, also avoiding and administering malaria treatment during pregnancy [10].

A study in Nigeria identified that ethnicity and high social class are significant predictors of parents seeking appropriate care for their children with febrile illness. The study also reviewed that mothers who attended secondary school education were less likely to seek appropriate care for their under-five children [11]. However, these local studies in Nigeria identified limited information about HSB of parents; HSB among parents is not attainable to national scale in view of the cultural diversity in Nigeria. Nevertheless, it is vital to obtain baseline information from parents on HSB of their children with febrile illness. Such information could provide awareness on interventions and education to improve parents HSB for children with febrile illness. Therefore, this study identified caregiver’s HSB for their children under-five years and the predictors of appropriate HSB among the caregivers.

## 2. Materials and Methods

This research was carried out in Imo state between June and September 2017. It was a cross sectional study, aimed to determine factors associated with HSB for fever treatment among caregivers of under-five children with fever. In the study area, South East region of Nigeria, which includes Imo State, indicated in Figure 1 map of Nigeria, it was reported that there was malaria prevalence of 27.6 % among children age 6 to 59 months [7]. A study was conducted in the region; however, the study focused on socio-demographic factors; therefore, this study was conducted in Imo State Nigeria to determine other factors that are associated with HSB of caregiver of under five children with fever in the region [12]. A two-proportion sample size formula was used [13]. Sample size was based on a previous cross-sectional study. In the formula, P1 = 0.594 and P2 = 0.419 [14]. After considering for design effect of 2 due to the multistage random sampling used and adjusting for 90% response rate, the minimum sample size required in this study is 559. Multistage sampling was employed. In the first stage, simple random sampling using a computer-generated table of random numbers to select four out of 27 local governments, and in the second stage, simple random sampling with probability proportionate to size was employed to select the number of households from each local government area (LGA). All households were used; inclusion and exclusion criteria were applied. The inclusion criteria were caregivers of children under the age of five years old who had fever two weeks prior to the study in Imo State Nigeria. Exclusion criteria were caregivers of children under the age of five years old with history of diagnosed long-term illnesses. They were excluded because care-seeking for these groups of children could be different compared to those children that did not have long-term illness. Households were selected from the list of households.

This study was approved by Ethics Committee for Research involving Human Subjects of University Putra Malaysia (JKEUPM). Permission approval letters were obtained from the local governments where the study was conducted in Nigeria. Upon explaining the objectives of the study, written consent was obtained from all respondents and confidentiality was maintained.

Data were obtained using a pretested structured self-administered questionnaire. Appropriate HSB was operationally defined as treatment sought from health facility within 24 h of onset of fever [15]. The questionnaire was pre-tested among 56 caregivers of under-five year old children, which is 10 percent of the total study population. The pretest was done in Orlu LGA, which is different from the selected study LGAs, but they have similar demographic backgrounds to the study population. With a Cronbach’s Alpha of 0.708, the questionnaire is considered valid.

In most parts of the world where malaria is hyper-endemic, parents may wish to take their children to healthcare facilities. However due to logistic reasons such as living in remote areas, no good roads, public transport, and distance to health facility, taking the child to health facility within 24 h might be difficult. The study location has only few hospitals and private clinics, around fifteen; St. Mary’s Joint Hospital, Amaigbo, is the only government hospital in the study location, and it is located in Nwangele LGA, which is not close to the other selected LGA, and this may affect the caregiver’s HSB. However, if they seek treatment from traditional healers, it is considered as inappropriate HSB. The main reason is because it is difficult to standardize traditional healer practices; it is uncertain whether the parents are seeking a trained and certified traditional healer or the untrained one, whose quality of care is questionable and therefore cannot be considered as giving appropriate treatment for malaria.

Only respondents that sought treatment for their under-five children with fever from a health facility within 24 h of fever onset were considered as caregivers with appropriate HSB. Statistical Package for Social Sciences (SPSS) version 22 was used to analyze all data and *p*-value was set as 0.05. Descriptive statistics including frequencies of all variables were obtained, and multiple logistic regression was used to determine the predictors of HSB.

## 3. Results

### 3.1. Socio-Demographic Characteristics of Respondents

A total of 559 eligible respondents were recruited. However, only 553 responses were analyzed in this study; six were excluded during analysis due to severe missing data of important information. This gives the response rate of 98.9%, which is much higher than the calculated response rate, 553 respondents is enough for this study.

Table 1 shows socio-demographic characteristics of the respondents. Among the 553 children of the caregivers in the study, there were 55.9% males and 44.1% females. A total of 51.9% of the children were below 27 months old. The result of this study indicates that caregivers’ age ranged from 15 to 64 years. Majority of the caregivers were mothers of the children (93.3%) and 86.8% were married. A total of 69.1% attended secondary school level; 60.8% were currently employed, and 40.3% of the caregivers earn between 51 to 111 US$ per month. A total of 57.3% of the respondents had more than four family members in their households. Majority of the caregivers were of Igbo ethnic group, (96.7%) and a total of 96.9% of the respondents live in the village.

### 3.2. Caregivers’ Health-Seeking Behavior 

Table 2 shows the distribution of respondent’s HSB. Among the 553 caregivers in this study, all caregivers sought treatment for their child’s fever. A total of 109 (19.7%) caregivers sought treatment from government hospital, 125 (22.6%) of the caregivers sought treatment from government health centre and (45.4%) sought treatment from medicine shop. Concerning time taken to treat the child’s fever, 53% of the caregivers sought treatment for the child within 24 h of fever onset, 44.8% sought treatment for the child two days after fever and 2.0% sought treatment for the child more than 2days of fever.

All the 553 caregivers gave medicine to their child during fever; however, 67.2% gave paracetamol to their child, 30.4% of the caregivers gave anti-malaria medicine to their child, and 2.4% gave herbs to their child. Regarding the days it takes for the caregivers to administer medicine to the child, 53.2% of the caregivers gave their child medication the same day of fever onset, 44.8% of the caregivers gave medication to their child 2 days of fever, and 2.0% of the caregivers gave their child medication more than 2 days after fever onset. In addition, 27.7% took their child for blood test during fever onset.

### 3.3. Characteristics of Appropriate and Inappropriate HSB

Table 3 shows the characteristics of appropriate and inappropriate HSB. Only caregivers who sought treatment from health facility (government hospital, government health centre, and private hospital only) within 24 h of onset of fever are considered as caregivers that sought appropriate HSB. Caregivers who sought treatment from non-health facility (medicine shop, traditional practitioner, and others) within 24 h of onset of fever and those who sought treatment in health facility after 24 h of onset of fever are considered as caregivers with inappropriate HSB. Among the 553-respondent recruited in the study, only 103 (18.6%) of the caregivers sought appropriate HSB and 450 (81.4%) of the caregivers sought inappropriate HSB.

### 3.4. Predictors of Appropriate Health Seeking Behavior 

Simple logistic regression was conducted first, before multiple logistic regression, in order to acquire the predicting factors influencing HSB. Table 4 shows the crude odd ratio for the factors affecting HSB. Variables that were statistically significant in simple logistic regression were gender of the child, age of the child, relationship to child, occupation, monthly income, number of household members, and decision making when seeking treatment for the child.

Multivariate logistic regression analysis was carried out on the variables that were significant at simple logistic regression at *p*-value of 0.05. Table 5 shows adjusted odd ratio (aOR) of predictors of HSB. Gender of the child remained a significant predictor of HSB. Male children had 2.7 times odds of seeking appropriate care compared to female children (aOR = 2.760; 95% CI: 1.536–4.958). Child’s age remained a significant predictor of HSB. Children who were less than 27 months were 2.8 times more likely to be taking for appropriate care compared to children more than 27 months old (aOR = 2.804; 95% CI: 1.485–5.295). Caregivers who were currently employed were 1.8 times more likely to practice appropriate HSB compared to caregivers that were currently not employed (aOR = 1.882; 95 %CI: 1.014–3.493). Number of household members was a predictor of HSB. Households that had less than four members in the family were 2.5 times more likely to practice appropriate HSB compared to caregivers that had more than four members in the family (aOR = 2.504; 95% CI: 1.464–4.283). Furthermore, decision in taking the child for treatment also remained a significant predictor of HSB. Caregivers who made decision to take the child for treatment at early stage of fever were more likely to seek appropriate HSB for the child (aOR = 0.142; 95% CI: 0.032–0.619) compared to caregivers who sought treatment for the child when illness was severe.

## 4. Discussion

This study was carried out to assess HSB for fever treatment among caregivers of under-five children with fever in Imo State Nigeria. The study revealed important findings about caregivers’ HSB for fever treatment among under-five children. Caregivers’ HSB was poor for fever cases; only 18.6% practiced appropriate HSB, which were caregivers that sought treatment from health facilities within 24 h of onset of fever for their under-five year’s children. The reason why the percentage of appropriate HSB was low in this study was because the majority of caregivers first sought treatment in a drug shop rather than health facility, and most caregivers that sought treatment in a health facility delayed fever onset more than 24 h. This could be because drug shops are mostly first access places to purchase healthcare products like drugs and other services [16]. A study reported that caregivers sought advice and diagnosis from medicine store, mainly people who lived in rural areas where health facilities are limited [17]. This result is similar to the result of a study conducted in Nigeria, among caregivers of under-five years old children, which reported that 20.5% of the caregivers sought treatment from a patent medicine dealer [12].

Caregivers’ practice of not using health facility and delay when seeking treatment is of extreme concern as only 18.6% of the caregivers sought treatment for their children from health facility within 24 h of fever onset. While 81.4% of the caregivers sought treatment for their under-five children from non-health facilities within 24 h of fever onset, caregivers who sought treatment from a health facility after 24 h of fever onset, which comprises medicine stores and traditional healers, were termed as caregivers with inappropriate HSB. The consequence of these behaviors to the children was that inappropriate HSB is a risk factor for illness complications [18]. Seeking treatment for a child with fever from an inappropriate site or treatment delay makes the child’s condition worse, with elevated fever, lethargy, loss of appetite, and poor feeding [3]. When these symptoms arise, it entails urgent medical care, and when treatment is delayed or the child not taken to health facility for treatment, it may lead to child mortality. Moreover, if treatment is initiated on time, it will help for better prognosis to the children [4]. However, the 18.6% of caregivers who sought appropriate treatment for their under-five children within 24 h of fever onset was far below the 80% of the set targets of national malaria control program [19]. The result of this study is comparable with a study in Myanmar which reported that only 35.3% of caregivers sought appropriate treatment [20]. Appropriate HSB of the caregivers in this study is low because most of caregivers prefer patronizing medicine shop rather than health facility. Caregiver’s behavior when seeking treatment for their children could improve; therefore, it is essential to improve caregiver’s awareness on the method of seeking early and appropriate treatment and increase positive attitude towards programs and initiatives that target eliminating malaria. Education campaigns are important for caregivers to seek treatment for their children with fever irrespective of other symptoms. It is vital to use appropriate preventive practice in households were there are under-five children, households would have two or more insecticide treated net (ITN).

This study revealed that gender of the child was a significant independent predictor of HSB. The odds of taken a child for appropriate care among male children was almost three times higher when compared to female children. This finding is corroborated with a study in Bangladesh which reported that male children were more likely to sought appropriate treatment compared to female children [21]. The possible reason could be because of the culture in the study area. In Nigeria society, there is a stereotype and biases with gender, in some communities’ male children have more priority than female children [22]. Male children are highly treasured in most Nigerian families when compare to female children [23]. A study conducted on HSB among children revealed that male children were more likely to be taken to health facility during illness, they were taken to private physicians or the caregiver’s purchase prescribed medication if the child is a male child compared to when the child is female child [24].

Age of the child was also a significant independent predictor of HSB in this study. This study revealed that younger children were more likely to be taken for appropriate treatment compare to older children. Study in Nigeria corroborates this study’s findings, which reported that younger children were more likely to seek appropriate treatment [25]. This study also coincides with the findings of a previous study among caregivers of under-five children in Bangladesh which reported that children younger than 12 months old were more likely to be taken to a trained healthcare provider compare to children of three years old [21]. This could be because individual and health system factors influence HSB [26]. This could also be because of caregiver’s perception that younger children are tender and require appropriate care compare to older children.

Occupation was a significant independent predictor of HSB. The result of this study revealed that caregivers who were currently employed sought appropriate care for their under-five children. This finding contrast with the report from a study in Nigeria [12]. The possible reason could be because, caregivers that were currently employed have their own income; they were empowered and were likely to seek appropriate treatment for their under-five children. It could also be because the caregivers that were currently employed, could afford the treatment of the child without waiting for the child’s father to provide the money for treatment so they can act fast by taking the child for appropriate care. Moreover, it could also be because occupation is link with appropriate HSB, when caregivers are employed, they have the ability to access resources as well as their economic power in their family which increase their power to make decision concerning the care of the child.

The result of this study shows that number of household members was a significant independent predictor of HSB. Household that had less than four members in the family were more likely to practice appropriate HSB compared to caregivers that had more than four members in the family. Study conducted in Myanmar contradicts the findings of this study which reported that number of household members was not a significant independent predictor of HSB [20]. Reason could be because when there is less family members, treatment will be initiated on time, compare to household where there is more members or more than one under-five child in the family and all the children become ill at the same time, it might be difficult to manage all of them at the same time.

Deciding factor when seeking treatment for the child was also a significant independent predictor of HSB. The finding of this study revealed that when treatment is decided at early stage of fever, the child is likely to be taken for appropriate care. This could be because the caregivers took the child for treatment when the illness is mild and since the fever was at early stage, caregivers may have time to choose where to take the child for treatment. The result contradicts with the report from a study in Myanmar [20].

## 5. Conclusions

Caregivers’ HSB for fever cases among their under-five year old children was poor; only 18.6% practiced appropriate HSB. Gender of child, age of child, caregivers’ occupation, number of household members, and decision making were predictors of appropriate HSB. It is essential to educate caregivers and emphasize on early treatment of fever and appropriate use of health facilities. Campaigns and programs to educate the caregivers on more of the dangers when treatment is delayed for their child could help to improve caregiver’s pattern of treatment seeking. These findings will be used to improve intervention at the community level and will be compared with follow-up data to evaluate its effectiveness.

## Figures and Tables

**Figure 1 ijerph-16-03752-f001:**
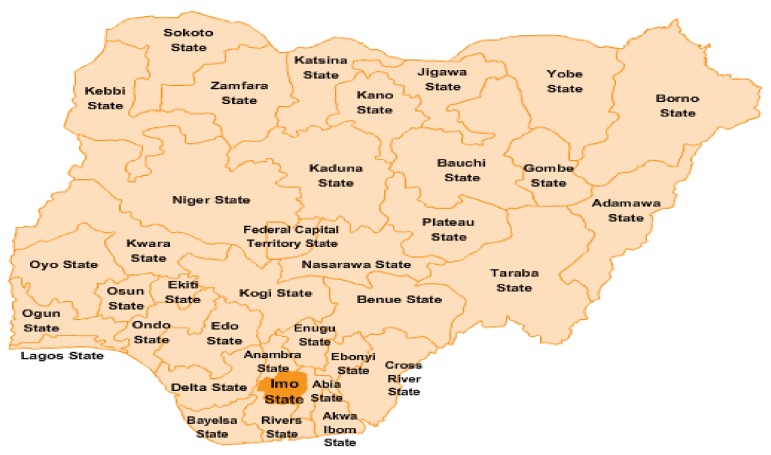
Map of Nigeria. (Source: google.com. Images for map of Nigeria)

**Table 1 ijerph-16-03752-t001:** Socio-demographic characteristics of respondents (N = 553).

Variables	Frequency (n)	Percentage (%)
Gender of Child Male Female	309 244	55.9 44.1
Age of Child in Months <27 >27	287 266	51.9 48.1
Age of Caregiver (Years) 15–24 25–34 35–44 45–54 55–64	174 214 94 70 1	31.5 38.7 17.0 12.7 0.2
Relationship to Child Mother Others	516 37	93.3 6.7
Marital Status Single Married Widow	38 480 35	6.9 86.8 6.3
Level of Education Primary Secondary University	6 382 165	1.1 69.1 29.8
Occupation Employed Unemployed	336 217	60.8 39.2
Household Monthly Income US$ 14–50 51–111 112–195 196 and above	36 223 205 89	6.5 40.3 37.1 16.1
Number of Household Members <4 >4	236 317	42.7 57.3
Ethnicity Igbo Ibibio Efik Yoruba	535 9 3 6	96.7 1.6 0.5 1.2
Place of Residence City Village	17 536	3.1 96.9

**Table 2 ijerph-16-03752-t002:** Caregivers’ health seeking behavior (N = 553).

Variables	Frequency (n)	Percentage (%)
Seek treatment for child fever Yes	553	100
Where treatment was first sought Government hospital Government health Centre Private hospital Medicine shop Traditional practitioner Others	109 125 25 251 16 27	19.7 22.6 4.5 45.4 2.9 4.9
Time Taken to Seek Treatment 1 day (24 h) 2 days (more than 24 h) Others	294 248 11	53.2 44.8 2.0
Any Medicine Given to Child during Fever Yes	553	100
What medicine was given to the child Paracetamol Anti-malaria medicine Herbs	372 168 13	67.2 30.4 2.4
Days Taken to Give Medicine to The Child Same day Next day Two days after fever onset	296 245 12	53.5 44.3 2.2
Blood Test Yes No	153 400	27.7 72.3

**Table 3 ijerph-16-03752-t003:** Characteristics of appropriate and inappropriate HSB.

Health Seeking Behavior	Place and Time of Treatment	N (%)
Appropriate	Treatment from health facility within 24 h of fever onset.	103 (18.6%)
Inappropriate	Treatment from non-health facility within 24 h of fever onset and treatment in health facility after 24 h of fever onset.	450 (81.4%)
Total		553

**Table 4 ijerph-16-03752-t004:** Simple logistic regression showing crude odd ratio (OR) of predictors of appropriate health seeking behavior (N = 553).

Factors	S. E	Wald Statistics	df	Crude OR (95% CI)	*p*-Value
Gender of Child Male Female	0.258	24.442	1	3.585 (2.161–5.947) 1.000	<0.001 *
Age of Child <27 ≥27	0.282	42.399	1	6.267 (3.607–10.889) 1.000	<0.001 *
Age of Caregivers 15–30 31–45 >45	0.409 0.409	1.910 3.099	1 1	0.568 (0.255–1.267) 0.487 (0.218–1.085) 1.000	0.209 0.167
Relationship to Child Mother others	0.735	3.882	1	4.259 (1.008–18.002) 1.000	0.049 *
Occupation Employed Unemployed	0.267	19.356	1	3.231 (1.916–5.449) 1.000	<0.001 *
Household Monthly Income US$ ≥115 <109	0.241	18.618	1	2.825 (1.825–4.529) 1.000	<0.001 *
Number of Household Member <4 ≥4	0.236	35.103	1	4.050 (2.550–6.434) 1.000	<0.001 *
Knowledge Score High Low	0.228	0.001	1	1.006(0.644–1.572) 1.000	0.978
Preventive Score High Low	0.236	2.244	1	0.702(0.443–1.115) 1.000	0.134
Decision Making Early stage of fever Severity of illness	0.723	14.161	1	0.066(0.016–0.272) 1.000	0.001 *

OR: odds ratio, CI: confidence interval, * significant at *p*-value ≤ 0.05, S.E: Standard error.

**Table 5 ijerph-16-03752-t005:** Multiple logistic regression showing adjusted odd ratio (AOR) of predictors of appropriate health seeking behavior.

Factors	B	S. E	Wald	df	Adjusted OR	95% CI	*p*-Value
Gender of Child Male Female	1.015	0.299	11.528	1	2.760 1.000	1.536–4.958	0.001 *
Age of Child (Months) <27 months ≥27 months	1.031	0.324	10.115	1	2.804 1.000	1.485–5.295	0.001 *
Occupation Employed Unemployed	0.632	0316	4.016	1	1.882 1.000	1.014–3.493	0.045 *
Number of Household Members <4 ≥4	0.918	0.274	11.238	1	2.504 1.000	1.464–4.283	0.001 *
Decision Making Early stage of fever Severity of illness	1.954	0.752	6.747	1	0.142 1.000	0.032–0.619	0.009 *

AOR: odds ratio, CI: confidence interval, * significant at *p*-value ≤ 0.05, S.E: Standard error.

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
