# Peer review of "Predictors of Health-Seeking Behavior for Fever Cases among Caregivers of Under-Five Children in Malaria-Endemic Area of Imo State, Nigeria"

_ijerph, 2019, doi:10.3390/ijerph16193752_

Round 1
Reviewer 1 Report
The authors of this study have done that and the study is therefore of interest, however several details need to be improved/explained before this study is suitable for publication.
Table 1.- Occupation: Authors should mention the working type (e.g. some of these are working in government job, some of these are doing agriculture and some of these are going seasonally outside for labor work which may affect the HSB). Obviously everyone will do work for living their life, how can we put anyone in non-working group. Most of the time author used the term fever in the manuscript but it is not clear whether it was malaria or not. Author should include the malaria status in appropriate HSB and Inappropriate HSB group. Author should also compare the outcomes of delayed treatment in appropriate HSB and inappropriate HSB. The availability of Health facility, Public transportation and distance may surely affect the caregiver's HSB so author should mention these things in methodolgy (e.g. how many and which type of health facility they have in their study area)Author Response
Reply to Reviewer 1 comments
Table 1.- Occupation: Authors should mention the working type (e.g. some of these are working in government job,
Thank you
The terminology of "working" and "not working” has been changed to "employed" and "unemployed in table 1, 2 & 4
Most of the time author used the term fever in the manuscript but it is not clear whether it was malaria or not.
Thank you,
This study did not include malaria status to determine appropriate HSB; the authors only included fever cases.
The availability of Health facility, Public transportation and distance may surely affect the caregiver's HSB so author should mention these things in methodology.
Thank you,
It has been included in lines 119 to line 125.
How many and which type of health facility they have in their study area
Thank you,
The study location has only few hospitals and private clinic, around fifteen, St. Mary’s Joint Hospital, Amaigbo is the only government hospital in the study location, it is located in Nwangela LGA which is not close to the other three selected LGA, and this may affect the caregiver’s. HSB. Please see line 122 to line 125.

Reviewer 2 Report
The manuscript submitted by Oluchi et al, entitled as “Predictors of Health Seeking Behavior for Malaria Treatment Among Caregivers of Under-five Children with Fever in Imo State, Nigeria” describes health-seeking behavior (HSB) of caregiver’s for under-five children with fever. Presented study concluded that HSB was poor for fever cases and recommend gender awareness.
It is important to have an early intervention in cases of malaria, preferably within 24h of the onset of illness. However, it also depends on the accuracy of malaria diagnosis. Therefore, it is premature to relate each case of fever onset with malaria. Presented study lacks a careful analysis of their subjects, etiology of the fever, outcome of the illness, and measures (if any) taken. Due to lack of the follow-up information, the study is not complete and thus derivation of any conclusion should be seen with caution. Authors had derived several conclusions that are not based on their study or not analyzed properly. Therefore, manuscript needs to be revised thoroughly. My comments below may be helpful in adding value to this manuscript.
Introduction section lacks detail of effectiveness of HSB in controlling malaria. Therefor, rationale and importance of the study are not conveyed clearly. Line 77, states minimum sample size required is 559, but authors recruited only 553 respondents. Need explanation for under sampling and also questions validity of statistical analysis. Line 82: exclusion criteria are not defined. Why study was conducted in Imo State is not clear. Also, show a map of world and locate the area studied, to give a geographical perspective. How questionnaire was pre-tested? What points were considered? Line 106 and table 1: Salary should be converted to an internationally recognized currency, e.g. US$ or Euro. Line 94-95: It seems ONLY criteria to include a child was fever. Authors did not provide details of cause of fever. As noted in my general comments, it is the biggest flaw of the study. Fever could be due to prior vaccination, which require use of paracetamol only. It had been the case for a number of patients. Also, for malaria accurate diagnosis is a big issue. Thus, conclusions drawn are digressing and are not helpful in malaria disease management. Table 4, should include cases where “Appropriate HSB” was practiced. By including all the cases, the analysis does not provide any logical conclusion. As authors emphasize that “appropriate HSB” is the key for disease management, than analysis of inappropriate HSB is misleading. Similarly, table 5 does not mention what kind of cases is being analyzed? Again, the sample size is too small (as noted above). Discussion section does not provide clear conclusion or recommendation. E.g. gender bias is not clear since data pertaining to subject's gender that received appropriate HSB is not provided. Lines 183 to 187, and 202 onwards are not supported by study.
Similarly, discussions of age of the child, occupation of the caregivers etc. are not supported. Authors are encouraged to re-analyze their data by focusing on children who received “appropriate HSB”. Also, sample size need to be appropriate.
Minor comments:
Manuscript need proof reading.
Line 21: “respnts” should be “respondents”
Line 267: R.A.M. name is repeated.
Author Response
Reply to Reviewer 2 comments
Introduction section lacks detail of effectiveness of HSB in controlling malaria.
Thank you,
Detail of effectiveness of HSB in controlling malaria, please see lines 54 to 58.
Appropriate HSB is important when seeking treatment for fever among under-five children; this could help for better prognosis, because treatment will be initiated early. As suggested by WHO in guidelines for the treatment of malaria early HSB is a critical behavior that is helpful for the achievement and sustainability of controlling malaria.
Minimum sample size required is 559, but authors recruited only 553 respondents. Need explanation for under sampling and also questions validity of statistical analysis.
Thank you,
A total of 559 eligible respondents were recruited. However, only 553 responses were analysed in this study, six were excluded during analysis due to severe missing data of important information. This gives the response rate of 98.9%, which is much higher than the calculated response rate, 553 respondents is enough for this study. Cronbach’s Alpha of 0.708, the questionnaire is valid. Please see lines 137 to 141.
Exclusion criteria are not defined.
Thank you,
Exclusion criteria has been included in line 103 to 106.
Exclusion criteria were caregivers of children under the age of five years old with history of diagnosed long-term illnesses was excluded. They were excluded because care-seeking for these groups of children could be different compared to those children that does not have long-term illness.
Why study was conducted in Imo State is not clear.
Thank you
Reason why study was conducted in Imo State has been stated in line 80 to 85. In the study area South East region of Nigeria, which includes Imo State, it was reported that there is malaria prevalence of 27.6 % among children age 6 to 59 months. A study was conducted in the region however their study focused on Socio-demographic factors, therefore this study was conducted in Imo State Nigeria to determine other factors that are associated with HSB of caregiver of under five children with fever in the region. Please see lines 88 to 93
Also, show a map of world and locate the area studied, to give a geographical perspective.
Thank you,
Map of Nigeria has been included in line 109 to 112.
How questionnaire was pre-tested? What points were considered?
Thank you,
The questionnaire was pre-tested among 56 caregivers of children under the age of five years, which is 10 percent of the total study population. Please see lines 116 to 119.
Line 106 and table 1: Salary should be converted to an internationally recognized currency, e.g. US$ or Euro.
Thank you,
Table 1: Salary be converted to an internationally recognized currency, e.g. US$, please see Table 1
Line 94-95: It seems ONLY criteria to include a child was fever
Thank you,
Criteria to include a child, was fever and if the child is under-five years.
Authors did not provide details of cause of fever
Thank you,
Authors did not collect data on cause of fever. We looked at their HSB, which includes place of treatment and time of treatment.
Also, for malaria accurate diagnosis is a big issue.
Thank you,
we did not include the diagnose if it was malaria or not , we just included any fever cases.
Table 5 does not mention what kind of cases is being analyzed?
Thank you
Table 5. analyzed multiple logistic regression showing adjusted odd ratio (AOR) of predictors of appropriate health seeking behavior.
Again, the sample size is too small (as noted above).
Thank you,
The sample size is enough because it actually exceeded the estimated calculated sample size.
Discussion section does not provide clear conclusion or recommendation. E.g. gender bias is not clear since data pertaining to subject's gender that received appropriate HSB is not provided.
Thank you,
Data pertaining to subject's gender that received appropriate HSB has been provided. Please see lines 248 to 251.

Reviewer 3 Report
The present study is very interesting and obtains data on HSB in an area of ​​Nigeria. I think the research has enough merit and interest to be published, but some points should be clarified:
1.-The authors should explain the reasons for choosing Imo State among the different possible territories in Nigeria
2.-The minimum sample size calculated was 559, but the authors recruited only 553 caregivers. The authors should explain the reason for not having reached the expected number of caregivers.
3.-The authors comment that ethnicity was a significant independent predictor of HSB in the study. Despite this, ethnicity cannot be taken into account when 96.7% belong to the Igbo ethnic group. Could this fact have introduced a confounding factor in the results?
Author Response
Reply to Reviewer 3 comments
1-The authors should explain the reasons for choosing Imo State among the different possible territories in Nigeria
Thank you
Reason why the study was conducted in Imo State has been stated in line 88 to 93. In the study area South East region of Nigeria, which includes Imo State, it was reported that there is malaria prevalence of 27.6 % among children age 6 to 59 months. A study was conducted in the region however their study focused on Socio-demographic factors, therefore this study was conducted in Imo State Nigeria to determine other factors that are associated with HSB of caregiver of under five children with fever in the region.
2-The minimum sample size calculated was 559, but the authors recruited only 553 caregivers. The authors should explain the reason for not having reached the expected number of caregivers.
Thank you
A total of 559 eligible respondents were recruited. However, only 553 responses were analysed in this study, six were excluded during analysis due to severe missing data of important information. This gives the response rate of 98.9%, which is much higher than the calculated response rate, 553 respondents is enough for this study. Cronbach’s Alpha of 0.708, the questionnaire is valid. Please see lines 138 to 141.
3.-The authors comment that ethnicity was a significant independent predictor of HSB in the study. Despite this, ethnicity cannot be taken into account when 96.7% belong to the Igbo ethnic group. Could this fact have introduced a confounding factor in the results?
Thank you,
Ethnicity has been removed from the significant independent predictor of HSB in the study.

Round 2
Reviewer 2 Report
Revised manuscript can be accepted for publication.